# H1 restricts euchromatin-associated methylation pathways from heterochromatic encroachment

C Jake Harris[1]*[†‡], Zhenhui Zhong[1†§], Lucia Ichino[1], Suhua Feng[1,2], Steven E Jacobsen[1,2,3]*

[1]Department of Molecular, Cell and Developmental Biology, University of California, Los Angeles, Los Angeles, United States; [2]Eli & Edythe Broad Center of Regenerative Medicine & Stem Cell Research, University of California, Los Angeles, Los Angeles, United States; [3]Howard Hughes Medical Institute, University of California, Los Angeles, Los Angeles, United States

*For correspondence:
cjh92@cam.ac.uk (CJH);
jacobsen@ucla.edu (SEJ)

[†]These authors contributed equally to this work

Present address: [‡]Department of Plant Sciences, University of Cambridge, Cambridge, United Kingdom; [§]Ministry of Education Key Laboratory for Bio-Resource and Eco-Environment College of Life Sciences, Sichuan University, Chengdu, China

Competing interest: The authors declare that no competing interests exist.

**Abstract** Silencing pathways prevent transposable element (TE) proliferation and help to maintain genome integrity through cell division. Silenced genomic regions can be classified as either euchromatic or heterochromatic, and are targeted by genetically separable epigenetic pathways. In plants, the RNA-directed DNA methylation (RdDM) pathway targets mostly euchromatic regions, while CMT DNA methyltransferases are mainly associated with heterochromatin. However, many epigenetic features - including DNA methylation patterning - are largely indistinguishable between these regions, so how the functional separation is maintained is unclear. The linker histone H1 is preferentially localized to heterochromatin and has been proposed to restrict RdDM from encroachment. To test this hypothesis, we followed RdDM genomic localization in an *h1* mutant by performing ChIP-seq on the largest subunit, NRPE1, of the central RdDM polymerase, Pol V. Loss of H1 resulted in NRPE1 enrichment predominantly in heterochromatic TEs. Increased NRPE1 binding was associated with increased chromatin accessibility in *h1*, suggesting that H1 restricts NRPE1 occupancy by compacting chromatin. However, RdDM occupancy did not impact H1 localization, demonstrating that H1 hierarchically restricts RdDM positioning. H1 mutants experience major symmetric (CG and CHG) DNA methylation gains, and by generating an *h1/nrpe1* double mutant, we demonstrate these gains are largely independent of RdDM. However, loss of NRPE1 occupancy from a subset of euchromatic regions in *h1* corresponded to the loss of methylation in all sequence contexts, while at ectopically bound heterochromatic loci, NRPE1 deposition correlated with increased methylation specifically in the CHH context. Additionally, we found that H1 similarly restricts the occupancy of the methylation reader, SUVH1, and polycomb-mediated H3K27me3. Together, the results support a model whereby H1 helps maintain the exclusivity of heterochromatin by preventing encroachment from other competing pathways.

## eLife assessment

This **important** study indicates a role for linker Histone H1 in protecting heterochromatic regions from certain types of repression. The experiments and data analysis that support the model for the role of linker Histone H1are **solid**, although additional experiments could provide a deeper mechanistic understanding. The study will be of broad interest to those interested in the role of chromatin in eukaryotic gene expression.

## Introduction

Eukaryotic genomes are compartmentalized into euchromatic and heterochromatin regions (*Ruiz-Velasco and Zaugg, 2017*). In euchromatin, nucleosomes are more accessible, while in heterochromatin nucleosomes are more compacted and restrictive to transcription. Protein coding genes are typically euchromatic, while non-coding and repetitive elements more often reside in heterochromatin. However, transposable elements, which are targeted for DNA methylation and silencing, are found in both euchromatic and heterochromatic regions of the genome (*Bourque et al., 2018*). The pathways responsible for deposition of methylation and silencing of TEs in these different regions are functionally and genetically distinct (*Du et al., 2015*).

In plants, euchromatic repetitive regions are targeted by the RNA-directed DNA methylation pathway (RdDM) (*Erdmann and Picard, 2020*), in which the concerted action of non-coding RNA polymerases and small RNAs direct the DNA methyltransferase, DRM2, to specific sites of the genome. Newly invading genetic elements are targeted by the 'non-canonical' RdDM pathway, and involve the action of Pol II derived small RNAs (*Cuerda-Gil and Slotkin, 2016*). Once established, the canonical RdDM pathway takes over, whereby small RNAs are generated by the plant-specific polymerase IV (Pol IV). In both cases, a second plant-specific polymerase, Pol V, is an essential downstream component. Pol V transcribes scaffold transcripts to which Argonaute-bound small RNAs can bind through base complementarity to recruit DRM2 to direct DNA methylation (*Wierzbicki et al., 2009*; *Wierzbicki et al., 2008*; *Zhong et al., 2014*). The recruitment of Pol V to chromatin is stabilized by interaction with the DNA methylation readers SUVH2 and SUVH9 (*Johnson et al., 2014*). However, methylation alone is not sufficient to explain Pol V occupancy, as Pol V is restricted to the edges of long TEs and euchromatic methylated regions, despite methylation levels being high in the bodies of long TEs and in heterochromatin (*Zhong et al., 2012*). Therefore, additional mechanisms must exist to regulate Pol V occupancy.

Heterochromatin is targeted for DNA methylation and silencing by CMT methyltransferases (*Stroud et al., 2014*; *Zemach et al., 2013*). CMT3 and CMT2 are responsible for CHG and CHH methylation maintenance, respectively, at these regions. The CMTs form an epigenetic feedback loop with the KYP family of H3K9me2 methyltransferases, thereby maintaining high levels of repression-associated non-CG methylation and H3K9me2 at heterochromatic regions of the genome (*Du et al., 2014*; *Du et al., 2012*; *Li et al., 2018*). As both DNA methylation and H3K9me2 are known to recruit components of the RdDM pathway (*Johnson et al., 2014*; *Law et al., 2013*), it is unclear how these regions remain functionally separated from RdDM.

The linker histone H1 associates with nucleosomes to modulate chromatin accessibility and higher order structures (*Saha and Dalal, 2021*). H1 binds to linker DNA as it exits from the nucleosome dyad and reduces linker DNA flexibility (*Bednar et al., 2017*). In plants, H1 is preferentially associated with heterochromatin, where it contributes to chromatin compaction (*Bourguet et al., 2021*; *Choi et al., 2020*; *Rutowicz et al., 2019*). Genetic evidence suggests that the chromatin remodeler DDM1 displaces H1 at heterochromatic TEs to facilitate access of CMT methyltransferases (*Zemach et al., 2013*), while biochemical data suggests that this effect could be indirect, via DDM1s interaction with the heterochromatin associated histone variant, H2A.W (*Osakabe et al., 2021*). Why DDM1 grants CMTs but not RdDM access to H1-containing nucleosomes is not clear. Recent structural data shows that nucleosome-bound DDM1 promotes DNA sliding (*Liu et al., 2024*), which may be sufficient for CMT to deposit methylation, but not for the RdDM machinery to become established. H1 also plays a key role in shaping nuclear architecture and preventing ectopic polycomb-mediated H3K27me3 deposition in telomeres (*Teano et al., 2023*).

Since H1 physically locks genomic DNA to the nucleosome and is preferentially associated with heterochromatin, it is a promising candidate for preventing RdDM encroachment into these regions. Recent evidence supports this hypothesis, with RdDM-associated small RNAs becoming more heterochromatically enriched in *h1* knockouts (*Choi et al., 2021*; *Papareddy et al., 2020*). However, small RNAs are not a direct readout of functional RdDM activity and Pol IV-dependent small RNAs are abundant in regions of the genome that do not require RdDM for methylation maintenance and that do not contain Pol V (*Stroud et al., 2014*). Here we directly tested whether RdDM occupancy is affected by loss of H1, by taking advantage of an endogenous Pol V antibody (which recognizes Pol V's largest subunit, NRPE1). We found that *h1* antagonizes NRPE1 occupancy throughout the genome, particularly in heterochromatic regions. This effect was not limited to RdDM, similarly impacting both

the methylation reader complex component, SUVH1 (*Harris et al., 2018*), and polycomb-mediated H3K27me3 (*Teano et al., 2023*). These results suggest that H1 acts in part to restrict the function of other epigenetic pathways.

## Results

To understand how H1 affects RdDM occupancy, we performed chromatin immunoprecipitation followed by sequencing (ChIP-seq) with a native antibody against NRPE1 (the largest catalytic subunit of Pol V, previously validated [*Liu et al., 2018*]). Two biological replicates of ChIP-seq in wild-type control Col-0 and in *h1.1–1/h1.2–1* double mutant (hereafter referred to as WT and *h1*, respectively) were performed. As previously reported (*Böhmdorfer et al., 2016*; *Liu et al., 2018*; *Zhong et al., 2012*), NRPE1 was enriched over short TEs and at the edges of long TEs in WT (*Figure 1A–B*). In the *h1* mutant, however, NRPE1 enrichment was markedly increased, with NRPE1 invading the heterochromatic bodies of long TEs (*Figure 1A–B*). The negative correlation between NRPE1 enrichment and TE length observed in WT was reverted in the *h1* mutant (*Figure 1C*). This suggests that mutation of *h1* facilitates the invasion of NRPE1 to more heterochromatic regions, in which long TEs tend to reside (*Bourguet et al., 2021*; *Zemach et al., 2013*). Consistent with this, NRPE1 increased more over heterochromatin associated CMT2 dependent hypo-CHH differentially methylated regions (DMRs) than at the RdDM associated DRM2 dependent CHH sites (*Figure 1D*). The preferential enrichment of NRPE1 in *h1* was more pronounced at TEs that overlapped with heterochromatin associated mark, H3K9me2 (*Figure 1E*). Comparing NRPE1 occupancy over TEs that had previously been classified by as either 'euchromatic' or 'heterochromatic' (based on a broad range of features and small RNA expression dynamics during embryonic development *Papareddy et al., 2020*), again, we found a striking increase in NRPE1 at heterochromatic over euchromatic TEs in *h1* (*Figure 1F*). From a chromosomal viewpoint, NRPE1 was preferentially enriched over pericentromeric regions in *h1*, and showed corresponding depletion from the more euchromatic chromosomal arms (*Figure 1G*, *Figure 1—figure supplement 1*). Importantly, we found no evidence for increased expression of NRPE1 or other methylation pathway components known to be involved in Pol V loading or stability in the *h1* mutant (*Supplementary file 1*). Together, the results indicate that H1 broadly restricts NRPE1 access to chromatin, and that this effect is more pronounced in heterochromatic regions of the genome. As H1 itself is preferentially localized to heterochromatin (see *Figure 1B, D and F*; *Bourguet et al., 2021*; *Choi et al., 2020*), this suggests that H1 directly antagonizes RdDM occupancy.

One mechanism by which H1 could restrict RdDM access is by promoting chromatin compaction. Consistent with this idea and previous reports (*Bourguet et al., 2021*; *Choi et al., 2020*; *He et al., 2019*; *Rutowicz et al., 2019*), we observed increased accessibility in heterochromatic regions in *h1*, which mirrored the preferential recruitment of NRPE1 to heterochromatin (*Figure 2—figure supplement 1A, B*). Loss of H1-mediated compaction in heterochromatin may be the indirect result of loss of heterochromatin-associated marks such as H3K9me2. To examine this possibility, we performed H3K9me2 ChIP-seq, and although we observed some minor reductions in H3K9me2 in *h1*, the losses were not preferentially in heterochromatin where we see the majority of NRPE1 gain (*Figure 2—figure supplement 1C, D*). While we cannot exclude the possibility that other heterochromatin-associated marks are altered, the data suggests that H1 primarily affects the accessibility, rather than the nature of chromatin in these heterochromatic regions.

To determine whether the antagonistic relationship between H1 and NRPE1 is reciprocal, we performed ChIP-seq using an endogenous antibody for H1, with two biological replicates in the WT and *nrpe1* mutant backgrounds. H1 levels were broadly unchanged in *nrpe1* at TE regions (*Figure 2A*). Similarly, over NRPE1-defined peaks (where NRPE1 occupancy is strongest in WT) we observed no change in H1 occupancy in *nrpe1* (*Figure 2B*). The results indicate that H1 does not invade RdDM regions in the *nrpe1* mutant background.

Next, we asked whether ectopic recruitment of RdDM may be sufficient to evict H1 from the genome. For this, we took advantage of the ZF-DMS3 transgenic lines that we previously characterized, in which zinc finger (ZF) fused DMS3 can recruit endogenous NRPE1 to >10,000 ectopic sites throughout the genome, corresponding to ZF 'off-target' binding sites (*Gallego-Bartolomé et al., 2019*). With these lines, we performed NRPE1 and H1 ChIP-seq in parallel. Compared to the non-transgenic control, we confirmed that NRPE1 was highly enriched at these 'off-target' regions (*Figure 2C*). However, with the same material, we observed no depletion of H1 at the ectopically



**Figure 1.** NRPE1 accumulates in heterochromatin in an *h1* mutant background. (**A**) Genome browser image showing increased NRPE1 enrichment in *h1* over heterochromatic (long, H3K9me2 enriched) and not euchromatic (short, not H3K9me2 enriched) transposable elements (TEs). (**B**) metaplot showing ChIP-seq enrichment of NRPE1 in wild-type (WT) vs. *h1*, and H1 occupancy in WT for reference (data from GSE122394), at short vs. long TEs. (**C**) boxplot showing association between NRPE1 and TE length in WT and *h1*. wilcoxon rank sum test p-values indicated. (**D**) As in (**B**) for *drm12* vs. *cmt2* hypo CHH differentially methylated regions (DMRs). (**E**) Violin plot inlaid with boxplot showing enrichment of NRPE1 at TEs that overlap with H3K9me2 peaks, vs. TEs that do not, in WT vs. *h1*. Boxplot medians are shown in blue. Wilcoxon rank sum test p-values indicated. (**F**) as in (**B**) at euchromatic vs. heterochromatic TEs. (**G**) Chromosomal plots showing NRPE1 enrichment in *h1* as compared to WT, with pericentromeric regions denoted in gray.

The online version of this article includes the following figure supplement(s) for figure 1:

**Figure supplement 1.** NRPE1 enrichment in wild-type (WT) vs *h1* when mapped to either TAIR10 or Col-CEN genome assemblies (*Naish et al., 2021*).



**Figure 2.** RNA-directed DNA methylation (RdDM) does not reciprocally affect H1 localization. (**A**) Scatter plot showing H1 enrichment in wild-type (WT) vs. *nrpe1* at euchromatic vs heterochromatic transposable elements (TEs). (**B**) Metaplot showing H1 enrichment at NRPE1 peaks in WT vs. *nrpe1* (**C**) H1 and NRPE1 occupancy at ZF-DMS3 off target peaks. Upper panel: H1 occupancy in WT vs. ZF108-DMS3. Lower panel: NRPE1 occupancy in WT vs. ZF108-DMS3. Chromatin from the same sample was used for both H1 and NRPE1 ChIP-seq data, so the lower panel serves as a control showing that ZF108-DMS3 is recruited to these off-target regions in these samples, despite H1 localization remaining unchanged. (**D**) Same as (**C**) at the subset of regions where CHH methylation is gained in ZF108-DMS3 (*Gallego-Bartolomé et al., 2019*), indicating that the fully functioning RdDM pathway is recruited to these regions.

The online version of this article includes the following figure supplement(s) for figure 2:

**Figure supplement 1.** Loss of H1 results in increased heterochromatic accessibility and NRPE1 redistribution.

**Figure supplement 2.** Benchmarking of H1 ChIP-seq.

bound NRPE1 sites (*Figure 2C*). We validated the quality of our H1 ChIPs, demonstrating antibody specificity, showing that they follow the previously described profiling patterns over H2A.W associated TEs (*Bourguet et al., 2021*), H3K27me3 marked/unmarked genes (*Teano et al., 2023*), and that H1.2 occupancy (*Teano et al., 2023*) is similarly enriched over our H1 peaks (*Figure 2—figure supplement 2*). To exclude the possibility that NRPE1 alone may be insufficient to evict H1, but that the fully functioning RdDM pathway may be required (which constitutes at least 32 described

proteins acting in concert [*Matzke and Mosher, 2014*]), we focused on the subset of ZF-DMS3 off-target sites that experience hyper-CHH methylation (*Gallego-Bartolomé et al., 2019*). The gain of ectopic methylation at these regions indicates that the full RdDM pathway is recruited and functions to deposit de novo methylation. However, even at these regions where H1 is mildly enriched in WT, we saw no evidence for H1 depletion in the ZF-DMS3 transgenic lines (*Figure 2D*). Together, these results indicate that H1 hierarchically and non-reciprocally restricts RdDM from overaccumulation in heterochromatin.

Given our observation that NRPE1 is redistributed in *h1*, we wondered whether this could help to explain some of the unusual methylation defects observed in the *h1* mutant. For instance, *h1* mutants experience contrasting loss and gain of DNA methylation at euchromatic and heterochromatic TEs, respectively (*Bourguet et al., 2021*; *Zemach et al., 2013*). Previous explanations to account for this suggested that the *h1* afforded increased accessibility of euchromatic TEs may promote increased access by activation-associated chromatin modifying enzymes, which in turn antagonize DNA methylation (*Zemach et al., 2013*). Another non-mutually exclusive scenario is that euchromatic loss and heterochromatic gain of methylation are caused by the relative shift of RdDM from euchromatin to heterochromatin. To directly test this possibility, we generated two independent NRPE1 CRISPR knockouts in the *h1* mutant background (*Figure 3—figure supplement 1A, B*), and performed whole genome bisulfite sequencing (WGBS) analysis, alongside *nrpe1*, *h1*, and WT controls. We confirmed *nrpe1* loss of function in the independent CRISPR KO lines by plotting CHH methylation over NRPE1 associated high confidence DMRs (*Zhang et al., 2018*), finding that CHH methylation is entirely abolished over these regions in both *nrpe1/h1* lines (*Figure 3—figure supplement 1C*). By analysis of chromosome level DNA methylation patterns in the different mutant backgrounds, we found that the striking pericentromeric increase in CG and CHG methylation observed in *h1*, is almost entirely independent of RdDM, as it was not rescued in *nrpe1/h1* (*Figure 3A*, *Figure 3—figure supplement 2*). In contrast, CHH methylation was mostly unaltered between the genotypes at the chromosome scale, with evidence for a slight reduction of CHH in the *h1/nrpe1* double mutant as compared to *h1* alone (*Figure 3A*, *Figure 3—figure supplement 2*).

Next, we compared methylation levels in euchromatic versus heterochromatic TEs. Heterochromatic TEs experienced major increases in CG and CHG methylation in *h1* as previously reported (*Zemach et al., 2013*), and these were independent of NRPE1 function, as the same increases were observed in *nrpe1/h1* (*Figure 3B*). CHH levels were largely unchanged at heterochromatic TEs in all mutant backgrounds (*Figure 3B*). At euchromatic TEs, CG and CHG methylation levels were largely unchanged in *h1*, while CHH levels were markedly reduced as previously reported (*Figure 3C*). CHH methylation levels were further reduced in *nrpe1*, consistent with the well-established role of RdDM in maintaining CHH at euchromatic TEs (*Stroud et al., 2013*), with the additional mutation of *h1* in *nrpe1/h1* having a minimal effect (*Figure 3C*). However, we noticed that the loss of CHG methylation in *nrpe1* was largely rescued in the *nrpe1/h1* double mutant (*Figure 3C*). This striking effect was specific to euchromatic TEs. NRPE1 is therefore required for CHG methylation maintenance at euchromatic TEs and in the absence of H1, loss of RdDM can be functionally compensated for by other methylation pathways. This compensation is likely due to the action of CMT3 (*Stroud et al., 2014*; *Zemach et al., 2013*). The effect is reminiscent of *h1*'s amelioration of methylation loss in the chromatin remodeler mutant, *ddm1* (*Zemach et al., 2013*), and further supports a role for H1 in demarcating the boundary between heterochromatic and euchromatic methylation pathways.

Calling DMRs in the mutant backgrounds, we found that *h1* hyper CHG DMRs were by far the most numerous as compared to any other context (*Figure 4A*). Consistent with the average methylation chromosomal and TE plots, these hyper CHG DMRs were highly enriched over pericentromeric heterochromatin (*Figure 4B*). To gain insight into the functional context of methylation in these regions, we performed an overlap analysis between *h1* hyper CHG DMRs and DMRs from 96 other *Arabidopsis* gene silencing mutants that were published previously (*Stroud et al., 2013*). Building similarity matrices based on pairwise overlapping scores (Co-Occurrence Statistics), we observed the separation of clustering of mutants belonging to different functional pathways (*Figure 4C*, *Figure 4—figure supplement 1*), consistent with previous studies (*Stroud et al., 2013*; *Zhang et al., 2018*). The *h1* hyper CHG DMRs clustered primarily with components of the CG maintenance pathway, including *met1* and *vim1/2/3* (*Figure 4D*). Interestingly, the *h1* hyper CHG DMRs also clustered and correlated strongly with *ddm1* hyper CHH DMRs. As heterochromatic transposons are known to transition from

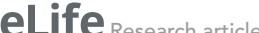

**Figure 3.** Major CG and CHG methylation gains in *h1* are independent of RNA-directed DNA methylation (RdDM). (**A**) Chromosomal view of DNA methylation levels (average of 10 kb windows) in the genotypes indicated on chromosome 1. Y-axis indicates fraction methylation (0–1). (**B**) Methylation level over heterochromatic transposable elements (TEs). Upper panel: methylation metaplots, Lower panel - kernel density plots. In the kernel density plots, the average methylation is calculated for each TE in both mutant and WT, then then the methylation difference is calculated. The plot shows the frequency density of TEs that gain/lose DNA methylation in the regions in the

*Figure 3 continued on next page*

*Figure 3 continued*

mutants indicated. (**C**) Methylation level over euchromatic TEs. Upper panel: metaplots, Lower panel - kernel density plots (as above). Arrow highlights the gain in CHG methylation in the *h1/nrpe1* double mutant, as compared to *nrpe1* alone.

The online version of this article includes the following figure supplement(s) for figure 3:

**Figure supplement 1.** NRPE1 CRISPR knock outs.

**Figure supplement 2.** Whole genome DNA methylation analysis in *h1* and *nrpe1* mutants showing individual biological replicates.

a state of quiescence to being actively targeted by RdDM in the *ddm1* mutant (*Panda et al., 2016*), this again suggests that H1 controls regions of the genome that are susceptible to spurious RdDM targeting.

While the majority of methylation changes observed in *h1* (hyper CG and CHG) were entirely independent of NRPE1, in the chromosomal plots we noted a subtle depletion of CHH methylation in the *h1/nrpe1* double mutant as compared to *h1* alone (*Figure 3A*), suggesting that redistribution of NRPE1 may have functional consequences on methylation patterning. To explicitly investigate these ectopically bound loci, we compared NRPE1 enrichment in WT versus *h1* and identified 15,075 peaks

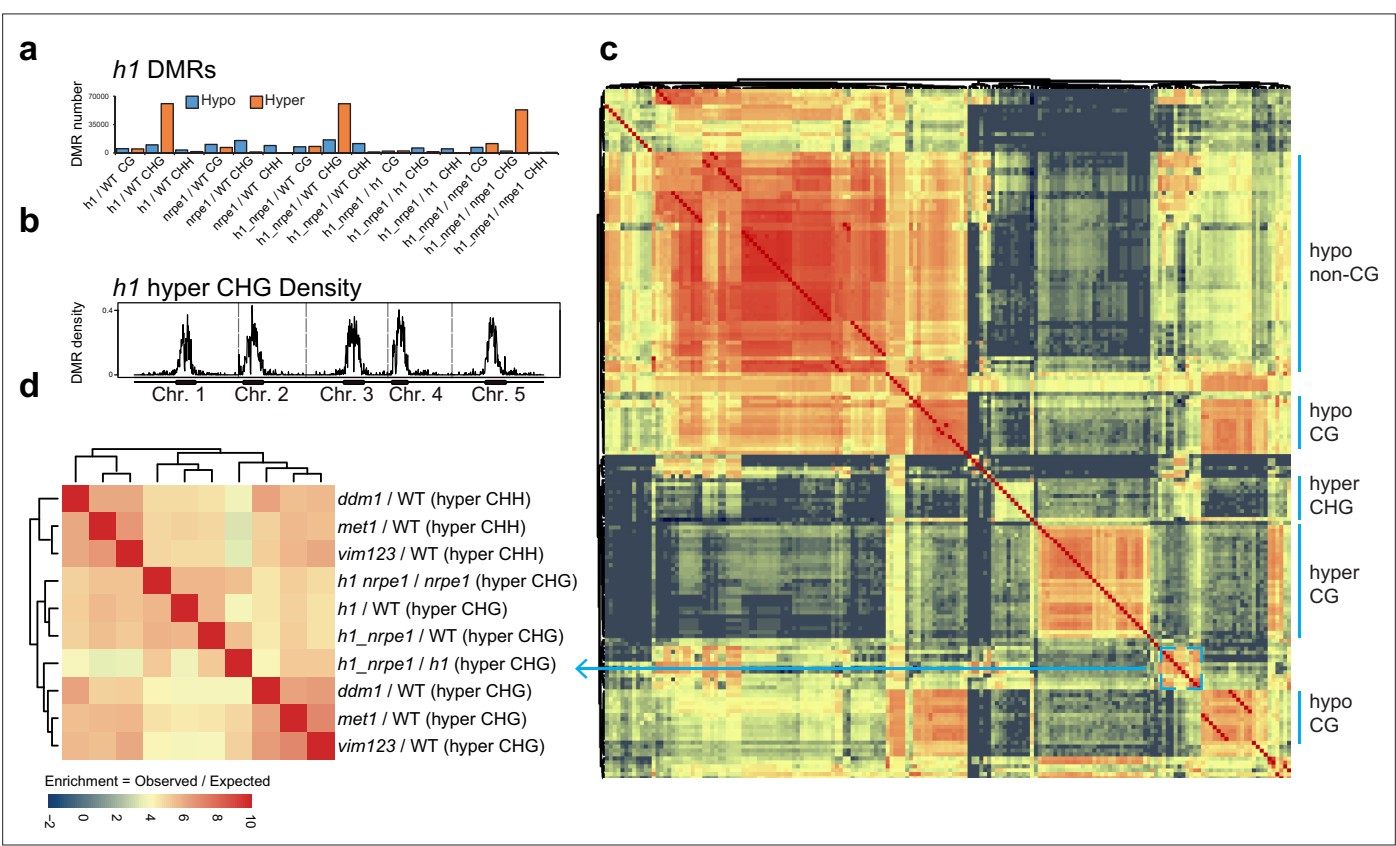

**Figure 4.** Overlap analysis of *h1* hyper CHG differentially methylated regions (DMRs) with 96 methylation mutants. (**A**) Number of hypo vs hyper DMRs in genotype comparisons indicated. (**B**) *h1* hyper CHG DMR frequency density plot over Chromosomes 1–5. (**C**) Similarity matrix based on pairwise overlapping scores (Co-Occurrence Statistics) of DMRs from 96 whole methylomes. The labels on the right summarise the major functional categories of the methylation mutant genotypes in the cluster block. (individual genotypes are shown in *Figure 4—figure supplement 1*). The color scale represents the number of observed DMR overlaps between the pairwise comparison indicated over the number of overlaps expected by chance (DMRs randomly distributed throughout the genome). Red means highly enriched overlapping DMRs (similar genomic distribution), blue means highly non-overlapping (different genomic distribution) (**D**) Zoomed in view of the cluster containing *h1* hyper CHG DMRs (**C**) (see blue box and arrow).

The online version of this article includes the following figure supplement(s) for figure 4:

**Figure supplement 1.** 96 mutant genotype comparison, as shown in *Figure 4*, with differentially methylated region (DMR) genotype comparison labels shown for visual inspection.



**Figure 5.** Patterns of NRPE1 re-localization in *h1* show corresponding methylation changes. (**A**) Methylation metaplot in the genotypes indicated over regions of the genome that gain NRPE1 in *h1*. Arrow highlights the change in average CHH methylation in the *h1/nrpe1* double mutant as compared to the single mutants. (**B**) Methylation metaplot of the genotypes indicated over regions of the genome that lose NRPE1 in *h1*.

The online version of this article includes the following figure supplement(s) for figure 5:

**Figure supplement 1.** Same data as shown in *Figure 5*, with individual biological replicates plotted separately.

**Figure supplement 2.** Genome browser images of representative regions that lose both NRPE1 occupancy and DNA methylation in *h1*.

that have significantly higher NRPE1 signal in the *h1* mutant. At these regions, CG and CHG methylation changes broadly mirrored that of heterochromatic TEs, again supporting the notion that NRPE1 primarily redistributes to heterochromatic regions of the genome in *h1* (*Figure 5A*, *Figure 5—figure supplement 1A*). However, we also observed a significant depletion of CHH methylation at these regions in the *nrpe1/h1* double mutant, as compared to either the *h1* and *nrpe1* alone, which was not observed when looking at heterochromatic TEs as a whole (compare *Figure 5A* to *Figure 3B*). This indicates that NRPE1 facilitates active deposition of CHH methylation at these newly bound locations. Reciprocally, we asked how methylation changes at regions of the genome that lose NRPE1 in *h1* (1,859 peaks), and we found that methylation levels are significantly depleted in all three contexts (CG, CHG, and CHH) in the *h1*, *nrpe1*, and *nrpe1/h1* double mutant backgrounds (*Figure 5B*, *Figure 5—figure supplements 1B and 2*). The mirrored loss and gain of methylation with changes in NRPE1 occupancy indicates that NRPE1 redistribution in *h1* directly impacts methylation patterning in cis.

The results thus far support a model whereby H1 prevents RdDM encroachment into heterochromatin. As H1 likely restricts access by reducing chromatin accessibility at these regions, we reasoned that this effect would not be RdDM specific and could affect other machinery in a similar manner. The

SUVH1 methyl binding protein directly binds CHH methylation in vitro, yet shows strong preferential recruitment to RdDM over CMT dependent sites in vivo (*Harris et al., 2018*). Along with SUVH3, SUVH1 functions to protect the expression of genes residing nearby to RdDM-targeted transposable elements by recruiting DNAJ1/2 transcriptional activators. We therefore wondered whether the SUVH1's occupancy might also be affected by H1. To test this, we performed ChIP-seq on SUVH1-3xFLAG in an *h1* mutant background (*Figure 6A*). As with NRPE1, SUVH1 was generally enriched throughout the genome in *h1*, but the effect was strongest over heterochromatin (long TEs, *cmt2*-dependent hypo-CHH DMRs, and heterochromatic TEs, *Figure 6B*). At the transcript level, we saw no evidence that SUVH1 or related complex components are upregulated in the *h1* mutant background (*Supplementary file 1*), however, we cannot rule out the possibility that the SUVH1-3xFLAG transgene is expressed more highly in *h1*. Teano et al recently showed that polycomb dependent H3K27me3 is redistributed in *h1*. We compared sites that gain NRPE1 to sites that gain H3K27me3 in *h1*, finding a statistically significant overlap (2.4 fold enrichment over expected, hypergeometric test p-value $2.1e^{-71}$). Reciprocally, sites that lose NRPE1 were significantly enriched for overlap with H3K27me3 loss regions (1.6 fold over expected, hypergeometric test p-value $1.4e^{-4}$). This indicates that RdDM and H3K27me3 patterning are similarly modulated by H1. To directly test this, we reanalyzed the H3K27me3 ChIP-seq data from Teano et al., finding coincident enrichment and depletion of H3K27me3 at sites that gain and lose NRPE1 in *h1* (*Figure 6E*). Therefore, H1 acts in a non-RdDM-specific manner, to prevent euchromatic machinery from heterochromatic encroachment, likely through promoting nucleosome compaction and restricting access to chromatin.

## Discussion

Here, we show that loss of H1 results in redistribution of RdDM from euchromatic to heterochromatic regions, consistent with recent findings from small RNA data (*Choi et al., 2021*; *Papareddy et al., 2020*). This ectopic accumulation results in modest but detectable NRPE1-dependent methylation, and it would be interesting to determine whether Pol V transcript production and DRM2 methyltransferase recruitment occur in a similar manner to euchromatin in these regions. It is important to note that NRPE1 binding was generally increased in the *h1* mutant (both euchromatin and heterochromatin), consistent with H1's widespread occupancy through the genome, but that the effect was strongest in heterochromatin where H1 shows maximal enrichment (*Bourguet et al., 2021*; *Choi et al., 2020*; *Rutowicz et al., 2019*). The antagonism of RdDM by H1 is non-reciprocal, as H1 levels were not affected by RdDM loss, or artificial gain at ectopic regions of the genome. H1-mediated restriction could be in part due to its structural capacity to reduce linker DNA flexibility (*Bednar et al., 2017*), thereby reducing polymerase access to unwind and interact with genomic DNA. Pol V requires the DDR complex for efficient association with chromatin (*Wongpalee et al., 2019*), so the presence of H1 could present an additional barrier to this complex. Mammalian H1 has been shown to compact chromatin in part through the phase-separating property of its highly charged intrinsically disordered domain (*Gibson et al., 2019*). Consistent with this, very recent work in *Arabidopsis* showed that H1 induced phase separation and that this capacity was essential for the stability of heterochromatic nuclear foci (*He et al., 2024*). Therefore, another possibility is that high H1 occupancy creates a biophysical environment that excludes Pol V and related complexes. Our data together with the observed heterochromatic encroachment of Pol IV-dependent small RNAs in *h1* (*Choi et al., 2021*; *Papareddy et al., 2020*), along with H1's recently described role in preventing spurious antisense transcription at protein-coding genes (*Choi et al., 2020*), support a polymerase indiscriminate model for H1's antagonism.

Small RNAs are relatively abundant in heterochromatin, despite Pol IV and RdDM being dispensable for methylation in these regions (*Stroud et al., 2014*). This suggests that Pol IV occupies heterochromatin, but that RdDM is either impotent for methylation deposition or acts redundantly with the CMT/KYP pathway. Here we detect NRPE1-dependent methylation at heterochromatic loci in *h1*, and combined with the finding that Pol V does not appreciably enter these regions in WT, suggests that H1 prevents RdDM from fully mobilizing in these regions by blocking Pol V. Recent evidence suggests that Pol V can be directly recruited by small RNAs to establish methylation (*Sigman et al., 2021*), and therefore mechanisms that preclude Pol V recruitment to heterochromatin would be particularly crucial in preventing redundant RdDM activity at regions where methylation has already been established.



**Figure 6.** SUVH1 and H3K27me3 encroaches heterochromatin in an *h1* mutant background. (**A**) Genome browser image showing SUVH1 re-localization in *h1* mirrors that of NRPE1. (**B–D**) Boxplot inlaid violin plots (boxplot medians shown in blue) showing SUVH1 enrichment in wild-type (WT) vs *h1* at short vs long transposable elements (TEs) (**B**), at euchromatic vs heterochromatic TEs (**C**) and at *drm12* vs *cmt2* hypo CHH differentially methylated regions (DMRs) (**D**). Wilcoxon rank sum test p-values indicated. (**E**) Metaplot of H3K27me3 dynamics in *h1* over regions that gain or lose NRPE1 in *h1*.

In *Arabidopsis thaliana*, RdDM targets evolutionarily young transposable elements (*Zhong et al., 2012*). H1 therefore helps to improve the efficiency of RdDM by focusing it on TEs with the greatest potential to mobilize. As transposons age, they eventually become targeted by CMTs. How this transition occurs is unclear, but our results indicate a potential role for the progressive accumulation of H1 in this process. Interestingly in tomato, CMTs rather than RdDM target the evolutionarily younger elements (*Wang and Baulcombe, 2020*), therefore it will be important to determine whether H1 plays a similar role in maintaining the barrier between methylation pathways in this context. Recent work has shown that H1 co-operates with H2A.W to promote compaction at heterochromatic regions (*Bourguet et al., 2021*). However, H2A.W was found to antagonize rather than promote H1 deposition at heterochromatin. Therefore, the features that drive H1 to preferentially incorporate into heterochromatin are yet to be discovered.

## Methods

### Plant materials and growth conditions

*Arabidopsis thaliana* plants in this study were Col-0 ecotype and were grown under 16 h light: 8 hr dark condition (on soil), or under constant light (on plates). The following plant materials were used: wild-type (WT, non-transgenic), *h1* (double homozygous mutant consisting of, *h1.1–1* [SALK_128430] and *h1.2–1* [GABI_406H11], originally described by *Zemach et al., 2013*), *nrpe1* (*nrpe1-11* [SALK_029919]), ZF-DMS3 (this is a homozygous transgenic line containing the pEG302-DMS3-3xFLAG-ZF construct in Col-0 WT, originally described by *Gallego-Bartolomé et al., 2019*), SUVH1-3xFLAG in *suvh1* (this is a complementing homozygous transgenic line containing the pEG302-gSUVH1-3xFLAG construct in *suvh1-1*, originally described by *Harris et al., 2018*), SUVH1-3xFLAG in *h1* (this line was generated for this work, by transforming the same pEG302-gSUVH1-3xFLAG construct into the *h1.1–1* [SALK_128430 ] and *h1.2–1* [GABI_406H11] double mutant background), and the *h1/nrpe1* CRISPR lines (described below).

### Generation of CRISPR lines

Two independent NRPE1 knockout lines were generated in the background of *h1* (*h1.1–1 h1.2–1* double, see above), designated as *h1/nrpe1* cKO_49 and *h1/nrpe1* cKO_63. The lines were generated using the previously published pYAO::hSpCas9 system (*Yan et al., 2015*) as described by *Ichino et al., 2021*. Briefly, the two guides (ATTCTTGACGGAGAGATTGT, and TCTGGCACTGAC AAACAGTT) targeting hSpCas9 to NRPE1 (AT2G40030) exons were sequentially cloned in the SpeI linearized pYAO::hSpCas9 construct by In-Fusion (Takara, cat #639650). The final vector was electroporated into AGL0 agrobacteria and transformed in *h1* mutant plants by agrobacterium-mediated floral dipping. T1 plants were selected on ½ MS agar plates with hygromycin B and were PCR genotyped to identify plants with deletions spanning the guide region. Selected lines were taken to T2 and genotyped to identify those that had segregated out the pYAO::hSpCas9 transgene. Cas9 negative plants that were homozygous for an NRPE1 deletion (*h1/nrpe1* cKO_49) or an indel-induced frameshift to premature stop codon (*h1/nrpe1* cKO_63) were confirmed in the T3 generation (see *Figure 3—figure supplement 1*).

### Chromatin immunoprecipitation sequencing (ChIP-seq) and western blot

All the materials for ChIP used in this paper were from pooled 10–12 days old seedlings grown on Murashinge and Skoog agar plates (1/2 X MS, 1.5% agar, pH5.7). The ChIP protocol used has been previously described (*Ichino et al., 2021*; *Villar and Kohler, 2018*), with minor modifications. Briefly, 2–4 g of seedling tissue was used for each sample. Samples were ground in liquid nitrogen and crosslinked for 10 or 12 min at room temperature in Nuclei Isolation buffer (50 mM Hepes pH8, 1 M sucrose, 5 mM KCl, 5 mM MgCl2, 0.6% triton X-100, 0.4 mM PMSF, 5 mM benzamidine hydrochloride, cOmplete EDTA-free Protease Inhibitor Cocktail (Roche)) containing 1% formaldehyde. The crosslinking reactions were stopped with 125 mM glycine by incubation at room temperature for 10 min. Crosslinked nuclei were filtered through one layer of miracloth, washed with Extraction Buffer 2, centrifuged through a layer of Extraction Buffer 3, and lysed with Nuclei lysis buffer (buffer compositions are described in the published protocol [*Villar and Kohler, 2018*]).

Chromatin was sheared using a Bioruptor Plus (Diagenode) (20 cycles of 30 s on and 30 s off) and immunoprecipitated overnight at 4 °C with an antibody. anti-NRPE1 (endogenous antibody, described in *Liu et al., 2018*), anti-H1 (Agrisera, AS11 1801), anti-H3K9me2 (abcam, 1220), anti-H3 (abcam, 1791), anti-FLAG (for both the SUVH1-3xFLAG and the DMS3-ZF ChIPs, we used Anti-FLAG M2 (Sigma, F1804)). 25 µl each of Protein A and Protein G magnetic Dynabeads (Invitrogen) were added to each sample and incubated for 2 more hours at 4 °C. The immunoprecipitated chromatin was washed with Low Salt (2 X), High Salt, LiCl, and TE buffers, for 5 min each at 4 °C (buffer compositions are described in the published protocol [*Villar and Kohler, 2018*]) and eluted twice with 250 µl of elution buffer (1% SDS and 0.1 M NaHCO3) in a thermomixer at 65 °C and 1000 rpm, 20 min for each elution. Reverse crosslinking was performed overnight at 65 °C in 0.2 M NaCl, and proteins were degraded by proteinase K treatment at 45 °C for 5 hr. The DNA fragments were purified using phenol:chloroform and ethanol precipitated overnight at –20 °C. Libraries were prepared using the Ovation Ultralow System V2 kit (NuGEN, 0344NB-A01) following the manufacturer's instructions, with 15 cycles of PCR. Libraries were sequenced on a HiSeq 4000 or NovaSeq 6000 instrument (Illumina) using either single- or paired-end 50 bp reads.

For the western blot shown in *Figure 2—figure supplement 2*, 100 ul of sheared chromatin was used as the input material for SDS-PAGE (as previously described [*Harris et al., 2016*]), using the anti-NRPE1 and anti-H1 antibodies described above.

### ChIP-seq data analysis

ChIP-seq data were aligned to the TAIR10 reference genome with Bowtie2 (v2.1.0) (*Langmead and Salzberg, 2012*) allowing only uniquely mapping reads with zero mismatch. Duplicated reads were removed by Samtools. ChIP-seq peaks were called by MACS2 (v2.1.1) and annotated with ChIPseeker (*Yu et al., 2015*). Bigwig tracks were generated using the deepTools (v2.5.1) (*Ramírez et al., 2016*) bamCompare function, with samples scaled by mapped read count (`--scaleFactorsMethod readCount`) prior to $log_2$ ratio or subtraction normalization against corresponding input (*Liu et al., 2018*) or control samples. These tracks were used to generate metaplots using the computeMatrix and plotProfile/plotHeatmap functions in deepTools. Differential peaks were called by the bdgdiff function in MACS2 (*Zhang et al., 2008*).

### Whole-genome bisulfite sequencing (BS-seq) library preparation

Whole genome bisulfite sequencing libraries were generated as previously described (*Harris et al., 2016*), with minor modifications. The DNeasy Plant Mini Kit (Qiagen #69104) was used to isolate genomic DNA. 150 µg of genomic DNA was used as input, a Covais S2 instrument was used for shearing (2 min), the Epitect Kit (QIAGEN #59104) was used for bisulfite conversion, and the Ultralow Methyl Kit (NuGEN) was used for library preparation and paired-end libraries were sequenced on a NovaSeq 6000 instrument.

### Whole-genome bisulfite sequencing (BS-seq) data analysis

Previously published whole-genome bisulfite sequencing data for mutants and wild type were reanalyzed from previous paper (*Stroud et al., 2013*). Briefly, Trim_galore (http://www.bioinformatics.babraham.ac.uk/projects/trim_galore/; *Krueger, 2019*) was used to trim adapters. BS-seq reads were aligned to TAIR10 reference genome by BSMAP (v2.90) and allowed two mismatches and 1 best hit (-v 2 w 1) (*Xi and Li, 2009*). Reads with three or more consecutive CHH sites were considered as unconverted reads and filtered. DNA methylation levels were defined as #mC/ (#mC + #unmC). DMR overlapping analysis were conducted by mergePeaks (-d 100) of Homer (*Heinz et al., 2010*) with WGBS data published previously (*Stroud et al., 2013*). The estimated conversion rates for all WGBS libraries are provided in *Supplementary file 2*.

## Acknowledgements

We thank Mahnaz Akhavan for support with high-throughput sequencing at the UCLA Broad Stem Cell Research Center BioSequencing Core Facility. CJH is supported by a Royal Society University Research Fellowship (URF\R1\201016). This work was supported by NIH R35 GM130272 to SEJ. SEJ is an Investigator of the Howard Hughes Medical Institute.

## Additional information

### Funding

| Funder | Grant reference number | Author |
|---|---|---|
| Royal Society | URF\R1\201016 | C Jake Harris |
| National Institutes of Health | GM130272 | Steven E Jacobsen |
| Howard Hughes Medical Institute | | Steven E Jacobsen |

The funders had no role in study design, data collection and interpretation, or the decision to submit the work for publication.

### Author contributions

C Jake Harris, Conceptualization, Data curation, Formal analysis, Investigation, Writing – original draft, Writing – review and editing; Zhenhui Zhong, Data curation, Formal analysis, Writing – original draft, Writing – review and editing; Lucia Ichino, Suhua Feng, Investigation; Steven E Jacobsen, Resources, Supervision, Funding acquisition, Writing – review and editing

### Author ORCIDs

C Jake Harris ⓘ https://orcid.org/0000-0001-5120-0377
Steven E Jacobsen ⓘ http://orcid.org/0000-0001-9483-138X

Reviewer #1 (Public review): https://doi.org/10.7554/eLife.89353.3.sa1
Reviewer #2 (Public review): https://doi.org/10.7554/eLife.89353.3.sa2
Author response https://doi.org/10.7554/eLife.89353.3.sa3

---

## Additional files

### Supplementary files

• Supplementary file 1. Differential expression analysis of RNA-seq data (from *Choi et al., 2020* PMID:31732458) for genes involved in DNA methylation, comparing wild-type (WT) to *h1* mutant genotypes.

• Supplementary file 2. Table showing bisulfite sequencing conversion efficiency estimation for the WGBS libraries generated and analysed in this paper.

• Supplementary file 3. List of .bed files used for analysis in this paper.

• MDAR checklist

### Data availability

Sequencing data have been deposited in GEO under accession code GSE225480.The pipelines and codes for downstream analysis are available on GitHub (https://github.com/Zhenhuiz/H1-restricts-euchromatin-associated-methylation-pathways-from-heterochromatic-encroachment, copy archived at *Zhenhuiz, 2023*). The bed files used to generate the plots are supplied in *Supplementary file 3*.

The following dataset was generated:

| Author(s) | Year | Dataset title | Dataset URL | Database and Identifier |
|---|---|---|---|---|
| Harris CJ, Zhong Z | 2023 | H1 restricts euchromatin-associated methylation pathways from heterochromatic encroachment | https://www.ncbi.nlm.nih.gov/geo/query/acc.cgi?acc=GSE225480 | NCBI Gene Expression Omnibus, GSE225480 |

The following previously published datasets were used:

| Author(s) | Year | Dataset title | Dataset URL | Database and Identifier |
|---|---|---|---|---|
| Choi J, Lyons DB, Kim MY | 2019 | DNA methylation and histone H1 jointly repress transposable elements and aberrant intragenic transcripts | https://www.ncbi.nlm.nih.gov/geo/query/acc.cgi?acc=GSE122394 | NCBI Gene Expression Omnibus, GSE122394 |
| Wolff L, Concia L, Biocanin I, Bourbousse C, Barneche F | 2023 | Linker histones mediate sequence-specific regulation of chromosomal organization and H3K27me3 enrichment over genes and telomeric repeats (ChIP-RX) | https://www.ncbi.nlm.nih.gov/geo/query/acc.cgi?acc=GSE160410 | NCBI Gene Expression Omnibus, GSE160410 |

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
