## [Editor Report · eLife assessment]

This **important** study indicates a role for linker Histone H1 in protecting heterochromatic regions from certain types of repression. The experiments and data analysis that support the model for the role of linker Histone H1are **solid**, although additional experiments could provide a deeper mechanistic understanding. The study will be of broad interest to those interested in the role of chromatin in eukaryotic gene expression.

---

## [Referee Report · Reviewer #1 (Public review)]

In this study, the authors obtained multiple, novel and compelling datasets to better understand the relationship between histone H1 and RNA-directed DNA methylation in plants. Most of the authors' claims concerning H1 and RNA polymerase V (Pol V) are backed by convincing and independent lines of evidence. However, Pol V produces noncoding transcripts that act as scaffold RNAs, which AGO4-bound siRNAs recognize in plant chromatin to mediate RNA-directed DNA methylation. Detection of Pol V transcript products at the sites of Pol V redistribution in h1 mutants would significantly enhance the impact of this manuscript. Below I have listed several strengths and a weakness of the manuscript.

Strengths:

- The authors report high-quality NRPE1 ChIP-seq data, allowing them to directly test how and where Pol V occupancy depends on histone H1 function in Arabidopsis.

- nrpe1 mutants generated via CRISPR/Cas9 in the h1 mutant background (nrpe1 h1.1-1 h1.2-1 triple mutants), allow the authors to study the role of Pol V in ectopic DNA methylation in H1-deficient plants.

- Pol V recruitment via ZincFinger-DMS3 expression (a modified version of Pol V's DMS3 recruitment factor) sends Pol V to new genomic loci and thus provides the authors with an innovative dataset for understanding H1 function at these sites.

Weakness:

- The manuscript does not include detection or quantification of Pol V transcripts generated at ectopic sites in the h1 mutant background. Pol V encroachment into heterochromatin in the h1 mutant is indirectly shown by NRPE1-dependent methylation at such ectopic sites.

Previous studies have charted the relationship between H1 function and RNA-directed DNA methylation (RdDM) via analyses of Pol IV-dependent 24 nt siRNAs and factors that recruit Pol IV (Choi et al., 2021 and Papareddy et al., 2020). Harris and colleagues have extended this work and shown that histone H1 function also antagonizes Pol V occupancy in the context of constitutive heterochromatin. The authors thus provide important evidence to show that H1 limits the encroachment of both polymerases Pol IV and Pol V into plant heterochromatin.

---

## [Referee Report · Reviewer #2 (Public review)]

Summary:

The main conclusion of the manuscript is that the presence of linker Histone H1 protects Arabidopsis pericentromeric heterochromatic regions and longer transposable elements from encroachment by other repressive pathways. The manuscript focuses on the RNA-dependent DNA-methylation (RdDM) pathway but indirectly finds that other pathways must also be ectopically enriched.

Strengths:

The authors present diverse sets of genomic data comparing Arabidopsis wild-type and h1 mutant background allowing an analysis of differential recruitment of RdDM component NPRE1, which is related to changes in DNA methylation and H1 coverage. The manuscript also contains recruitment data for SUVH1 in wild-type and h1 mutant backgrounds.

Furthermore, the authors make use of a line that recruits NRPE1 ectopically to show that H1 occupancy is not altered because of this recruitment. These data clearly show that there is a hierarchy in which DNA-methylation is impacted by presence of H1 while H1 distribution is independent of DNA-methylation.

Weaknesses:

The manuscript is driven by a strong and reasonable hypothesis that absence of H1 results increased access of chromatin binding factors and that this explains how the RdDM machinery is restricted from encroaching heterochromatic regions, which are particularly enriched in H1. Indeed, increased binding of NPRE1 at pericentromeric sites is observed; however, the major DNA-methylation changes at these sites are symmetric and not related to the RdDM pathway. Thus, the authors propose that many factors redistribute, which is again reasonable. The authors show redistribution of SUVH1 and relate their data to a previous report showing redistribution of the PcG machinery in H1 depletion mutants (Teano et al. in Cell reports Volume 42, Issue 8, 29 August 2023), but the manuscript provides limited mechanistic insight as to why there is a strong increase in heterochromatin symmetric DNA-methylation.

---

## [Author Response]

The following is the authors’ response to the original reviews.

**Recommendations for the authors:**

**Reviewer #1 (Recommendations For The Authors):**
Pg. 3 - lines 51-53: "Once established, the canonical RdDM pathway takes over, whereby small RNAs are generated by the plant-specific polymerase IV (Pol IV). In both cases, a second plant-specific polymerase, Pol V, is an essential downstream component." The authors' intro omits an important aspect of Pol V's function in RdDM, which is quite relevant to their study. Pol V transcribes DNA to synthesize noncoding RNA scaffolds, to which AGO4-bound 24 nt siRNAs are thought to base pair, leading to DRM2 recruitment for cytosine methylation near to these nascent Pol V transcripts (Wierzbicki et al 2008 Cell; Wierzbicki et al. 2009 Nat Genet). I recommend that the authors cite these key studies.

These citations have now been added (see line 57).

The authors provide compelling evidence that Pol V redistributes to ectopic heterochromatin regions in h1 mutants (e.g., Fig1a browser shot). Presumably, this would allow Pol V to transcribe these regions in h1 mutants, whereas it could not transcribe them in WT plants. Have the authors detected and/or quantified Pol V transcripts in the h1 mutant compared to WT plants at the sites of Pol V redistribution (detected via NRPE1 ChIP)?

Robust detection of Pol V transcripts can be experimentally challenging, and instead we quantify and detect NRPE1 dependent methylation at these regions (Fig 5), which occurs downstream of Pol V transcript production. However, we note detecting Pol V transcripts as a potential future direction in the discussion (see line 263).

Pg. 5 - lines 101-102: Figure 1e - "The preferential enrichment of NRPE1 in h1 was more pronounced at TEs that overlapped with heterochromatin associated mark, H3K9me2 (Fig. 1e). Was a statistical test performed to determine that the overall differences are significant only at TE sites with H3K9me2? Can the sites without H3K9me2 also be differentiated statistically?

Yes, there is a statistically significant difference between WT and h1 at both the H3K9me2 marked and unmarked TEs (Wilcoxon rank sum tests, see updated Fig 1e). The size of the effect is larger for the H3K9me2 marked TEs (median difference of 0.41 vs 0.16). Median values have now been added to the boxplots so that this is directly viewable to the reader (Fig 1e). This reflects the general increase in NRPE1 occupancy in *h1* mutants through the genome, with the effect consistently stronger in heterochromatin. In our initial version of the manuscript, we summarise the effect as follows “We found that h1 antagonizes NRPE1 occupancy throughout the genome, particularly at heterochromatic regions” (previous version line 83, current version line 95). Although important exceptions exist (see Fig 5, NRPE1 and DNA methylation loss in h1), we now make this point even more explicit, and have updated the manuscript at several locations (abstract line 26, results line 245, discussion line 265).

Pg. 5 - lines 108-110: The authors state, "Importantly, we found no evidence for increased NRPE1 expression at the mRNA or protein level in the h1 mutant (Suppl. Fig. 2)." But the authors did observe reduced NRPE1 transcript levels in h1 mutants, in their re-analysis of RNA-seq data and reduced NRPE1 protein signals via western blot in (Suppl. Fig. 2), which should be reported here in the results.

As described further below, we reanalysed *h1* RNA-seq from scratch, and see no evidence for significant differential gene expression of NRPE1. This table and analysis are now provided in Supplementary Table 1.

More importantly, the above logic about NRPE1 expression in h1 mutants assumes that NRPE1 is the stoichiometrically limiting subunit for Pol V assembly and function in vivo, but this is not known to be the case:(1) While NRPE1's expression is somewhat reduced (and not increased) in h1 mutant plants, we cannot be certain that other genes influencing Pol V stability or recruitment are unaffected by h1 mutants. I thus recommend that the authors perform RT-qPCR directly on the WT and h1 mutant materials used in their current study, quantifying NRPE1, NRPE2, NRPE5, DRD1, DMS3, RDM1, SUVH2 and SUVH9 transcript levels.(2) Normalizations used to compare samples should be included with RT-qPCR and western assays. An appropriate house-keeping gene like Actin2 or Ubiquitin could be used to normalize the RT-qPCR. Protein sample loading in Suppl. Fig. 2 could be checked by Coomassie staining and/or an antibody detection of a house-keeping protein.

We have now included a full re-analysis of *h1* RNA-seq (data from Choi et al 2020) focusing on transcriptional changes of DNA methylation machinery genes in the *h1* mutant. Of the 61 genes analysed, only AGO6 and AGO9 were found to be differentially expressed (2-3 fold upregulation). This analysis is now included as a table

(Supplementary Table 1). The western blot has been moved to Supplementary Fig 3 to now illustrate antibody specificity and H1 loss in the *h1* mutant lines, so NRPE1 itself serves as a loading control (Supplementary Fig 3a).

Pg. 6 - lines 129-131: The authors state that "over NRPE1 defined peaks (where NRPE1 occupancy is strongest in WT) we observed no change in H1 occupancy in nrpe1 (Fig 2b). The results indicate that H1 does not invade RdDM regions in the nrpe1 mutant background." This conclusion assumes that the author's H1 ChIP is successfully detecting H1 occupancy. However, in Fig 2d there does not appear to be H1 enrichment or peaks as visualized across the 10766 ZF-DMS3 off-target loci, or even at the selected 451 ZFDMS3 off-target hyper DMRs, where the putative signal for H1 enrichment on the metaplot center is extremely weak/non-existent.As a reference for H1 enrichment in chromatin (e.g., looking where H2A.W antagonizes H1 occupancy) one can compare analyses in Bourguet et al (2021) Nat Commun, involving co-authors of the current study. Bourguet et al (2021) Fig 5b show a metaplot of H1 levels centered on H2A.W peaks with H1 ChIP signal clearly tapering away from the metaplot center point peak. To my eye, the H1 ChIP metaplots for ZF-DMS3 offtarget loci in the current manuscript (Fig 2d) resemble "shuffled peaks" controls like those in Fig 5b of Bourguet et al (2021).Can one definitively interpret Fig 2d as showing RdDM "not reciprocally affecting H1 localization" without first showing the specificity of the ChIP-seq results in a genotype where H1 occupancy changes? Alternatively, could this dataset be displayed with Deeptools heatmaps to strengthen the evidence that the authors are detecting H1 occupancy/enrichment genome-wide, before diving into WT/nrpe1 mutant analysis at ZF-DMS3 off-target loci?

This is an excellent suggestion from the reviewer. We have now included several analyses that assess and demonstrate the quality of our H1 ChIP-seq profiles. First, as suggested by the reviewer, we show that our H1 profiles peak over H2A.W enriched euchromatic TEs as defined by Bourguet et al, mirroring these published findings. Next, we investigated whether our H1 profiles match Teano’s recently described pattern over genes, confirming a similar pattern with 3’ enrichment of H1 over H3K27me3 unmarked genes. Furthermore, we show that the H1 peaks defined here are similarly enriched with GFP tagged H1.2 from the Teano et al. 2023 study. These analyses that validate the quality of our H1 ChIP-seq datasets and bolster the conclusion that NRPE1 redistribution does not affect H1 occupancy. These new analysis are now presented in Supplementary Figure 3 and see line 153.

Pg. 8 - lines 228-230: The authors state that, "As with NRPE1, SUVH1 increased in the h1 background significantly more in heterochromatin, with preferential enrichment over long TEs, cmt2 dependent hypo CHH DMRs, and heterochromatic TEs (Fig. 6b)."Contrary to the above statement, the violin plots in Fig. 6c show SUVH1 occupancy increasing at euchromatic TEs in the h1 mutant. What statistical test allowed the authors to determine that the increase in h1 occurs "significantly more in heterochromatin"? The authors should critically interpret Fig. 6c and 6d, which are not currently referenced in the results section. More support is needed for the claim that SUVH1 specifically encroaches into heterochromatin in the h1 mutant, rather than just TEs generally (euchromatic and heterochromatic alike).

Similar to what we see for NRPE1, statistical tests that we have now performed show that SUVH1 is significantly enriched in h1 in all classes. Importantly however, the effect size is larger in all of the heterochromatin associated classes. We display these statistical tests and the median values on the plots so that effects are immediately viewable (see updated Fig 6).

In addition, the authors should verify that SUVH1-3xFLAG transgenes (in the WT and h1 mutant backgrounds, respectively) and endogenous Arabidopsis genes encoding the transcriptional activator complex (SUVH1-SUVH3-DNAJ1-DNAJ2) are not overexpressed in the h1 mutant vs. WT. Higher expression of SUVH1 or limiting factors in the larger complex could explain the observation of increased SUVH1 occupancy in the h1 background.

We do not see a difference in SUVH1/3/DNAJ1/2 complex gene expression in the *h1* background (see Supplementary Table 1). However, we cannot rule out that that our SUVH1-FLAG line in *h1* is more highly expressed than the corresponding SUVH1-FLAG line in WT. We now note this point in line 248.

Pg. 8 - lines 231-232: Here the authors make a sweeping conclusion about H1 demarcating, "the boundary between euchromatic and heterochromatic methylation pathways, likely through promoting nucleosome compaction and restricting heterochromatin access." I do not see how a H1 boundary between euchromatic and heterochromatic methylation pathways is revealed based on the SUVH1-3xFLAG occupancy data, which shows increased enrichment at every category interrogated in the h1 mutant (Fig 6b,c,d) and all along the baseline too in the h1 mutant browser tracks (Fig 6a). Can the authors provide more examples of this phenomenon (similar to Fig 6a) and better explain why their SUVH1-3xFLAG ChIP supports this demarcation model?

The general conclusion from SUVH1 about H1’s agnostic role in preventing heterochromatin access is now further supported from our findings with H3K27me3 (see Figure 6e and description from line 250). However, we agree that the demarcation model as initially presented was overly simplistic. This point was also raised by reviewer 2. We have removed the line highlighted by the reviewer in the revised version of the manuscript. In the revised version we clarify that H1 impedes RdDM and associated machinery throughout the genome (consistent with H1’s established broad occupancy across the genome) but this effect is most pronounced in heterochromatin, corresponding to maximal H1 occupancy (abstract line 26, results line 245, discussion line 265).

Corrections:Pg. 8 - lines 226-227: "We therefore wondered whether complex's occupancy might also be affected by H1." The sentence contains a typo, where I assume the authors mean to refer to occupancy by the SUVH1-SUVH3-DNAJ1-DNAJ2 transcriptional activator complex. This needs to be specified more clearly.

The paragraph has been updated (see from line 237).

Pg. 13 - lines 393-405: There are minor errors in the capitalization of titles and author initials in the References. I recommend that the authors proofread all the references to eliminate these issues:

Thank you, these have been corrected.

Choi J, Lyons DB, Zilberman D. 2021. Histone H1 prevents non-cg methylation-mediated small RNA biogenesis in arabidopsis heterochromatin. Elife 10:1-24. doi:10.7554/eLife.72676 (...)

Du J, Johnson LM, Groth M, Feng S, Hale CJ, Li S, Vashisht A a., Gallego-Bartolome J, Wohlschlegel J a., Patel DJ, Jacobsen SE. 2014. Mechanism of DNA methylation-directed histone methylation by KRYPTONITE. Mol Cell 55:495-504. doi:10.1016/j.molcel.2014.06.009 (...)

Du J, Zhong X, Bernatavichute Y V, Stroud H, Feng S, Caro E, Vashisht A a, Terragni J, Chin HG, Tu A, Hetzel J, Wohlschlegel J a, Pradhan S, Patel DJ, Jacobsen SE. 2012. Dual binding of chromomethylase domains to H3K9me2-containing nucleosomes directs DNA methylation in plants. Cell 151:167-80. doi:10.1016/j.cell.2012.07.034

**Reviewer #2 (Recommendations For The Authors):**
As for a normal review, here are our major and minor points.Major:(1) Lines 38 to 45 of the introduction are important for the subsequent definition of heterochromatic and non-heterochromatic transposons, but the definition is ambiguous. Is heterochromatin defined by surrounding context such as pericentromeric position or is this an autonomous definition? Can a TE with the chromosomal arms be considered heterochromatic provided that it is long enough and recruits the right machinery? These cases should be more explicitly introduced. Ideally, a supplemental dataset should provide a key to the categories, genomic locations and overlapping TEs as they were used in this analysis, even if some of the categories were taken from another study.

We have now added all the regions used for analysis in this study to Supplementary Table 3.

(2) Line 80: This would be the first chance to cite Teno et al. and the "encroachment" of

PcG complexes to TEs in H1 mutants

Done - “H1 also plays a key role in shaping nuclear architecture and preventing ectopic polycomb-mediated H3K27me3 deposition in telomeres (Teano et al., 2023).” See line 83

(3) It is "only" a supplemental figure but S2 but it should still follow the rules: Indicate the number of biological replicates for the RNA-seq data, and perform a statistical test. In case of WB data, provide a loading control.

We are now using the western blot to illustrate antibody specificity and H1 loss in the *h1* mutant lines, so NRPE1 itself serves as a loading control (Supplementary Fig 3a). For NRPE1 mRNA expression, we have now replaced this with a more comprehensive transcriptome analysis of methylation machinery in *h1* (see Supplementary Table 1).

(4) Lines 115 to 124 and corresponding data: Here, the goal is to exclude other changes to heterochromatin structure other than "increased access" in H1 mutants; however, only one feature, H3K9me2, is tested. Testing this one mark does not necessarily prove that the nature of the chromatin does not change, e.g. H2A.W could be differently redistributed, DDM1 may change, VIM protein, and others. Either more comprehensive testing for heterochromatin markers should be performed, or the conclusions moderated.

We have moderated the text accordingly (see line 135).

(5) Lines 166ff and Figure 1, a bit out of order also Figure 5: The general hypothesis is that NRPE1 redistributes to heterochromatic regions in h1 mutants (as do other chromatin modifiers), but the data seem to only support a higher occurrence at target sites.a. The way the NRPE1 data is displayed makes it seem like there is much more NRPE1 in the h1 samples, even at peaks that should not be recruiting more as they do not represent "long" TEs. It would be good to present more gbrowse shots of all peak classes.

We now clarify that *h1* does result in a general increase of NRPE1 throughout the genome, but the effect is strongest at heterochromatin. In our initial version of the manuscript, we summarise the effect as follows “We found that h1 antagonizes NRPE1 occupancy throughout the genome, particularly at heterochromatic regions” (previous version line 83, current version line 95). We have modified the language at several locations throughout the manuscript to make this point more clearly (abstract line 26, results line 245, discussion line 265). We include several browser shots in Supp Fig. 8.

b. The data are "normalized" how exactly?c. One argument of observing "gaining" and "losing" peaks is that there is redistribution of NRPE1 from euchromatic to heterochromatic sites. There should be an analysis and figure to corroborate the point (e.g. by comparing FRIP values). Figure 1b shows lower NRPE1 signals at the TE flanking regions. This could reflect a redistribution or a flawed normalization procedure.

The data are normalised using a standardised pipeline by log2 fold change over input, after scaling each sample by mapped read depth using the bamCompare function in deepTools. This is now described in detail in the Materials and Methods line 365, with full code and pipelines available from GitHub (https://github.com/Zhenhuiz/H1-restrictseuchromatin-associated-methylation-pathways-from-heterochromatic-encroachment).

d. Figure 1d and f show similar profiles comparing "long" and "short" TEs or "CMT2 dependent hypo-CHH" and "DRM2 dependent CHH". How do these categories relate to each other, how many fragments are redundant?

The short vs long TEs were defined in Liu et al 2018 (doi: 10.1038/s41477-017-0100-y) and the DMRs were defined in Zhang et al. 2018 (DOI: 10.1073/pnas.1716300115). There is likely to be some degree of overlap between the categories, but numbers are very different (short TEs (n=820), long TEs (n=155), drm2 DMRs (n=5534), CMT (n=21784)) indicating that the different categories are informative. We have now listed all the regions used for analysis in this study as in Supplementary Table 3.

e. The purpose of the data presented in Figure 1 b is to compare changes of NRPE1 association in H3K9me3 non-overlapping and overlapping TEs between wild-type and background, yet the figure splits the categories in two subpanels and does neither provide a fold-change number nor a statistical test of the comparison. As before, the figure does not really support the idea that NPRE1 somehow redistribute from its "normal" sites towards heterochromatin as both TE classes seem to show higher NRPE1 binding in h1 mutants.

There is a statistically significant difference between WT and *h1* at both the H3K9me2 marked and unmarked TEs, however, the size of the effect is larger for the H3K9me2 marked TEs (median difference of 0.41 vs 0.16). Median values have now been added to the boxplots so that this is directly viewable to the reader (Fig 1e). Although important exceptions exist (see Fig 5 – regions that lose NRPE1 and DNA methylation), this reflects the general increase in NRPE1 occupancy in *h1* mutants throughput the genome, with a consistently stronger effect in heterochromatin. As noted above, we have updated the manuscript to make this point more clearly (abstract line 26, results line 245, discussion line 265).

f. Panel g is the only attempt to corroborate the redistribution towards heterochromatic regions, but at this scale, the apparent reduction of binding in the chromosome arms may be driven by off-peak differences and normalization problems between different ChIP samples with different signal-to-noise-ratio.

We describe our normalisation and informatic pipeline in more detail in the Materials and Methods line 365. It is also important to note that the reduction is not only observed at the chromosomal level, but also at specific sites. We called differential peaks between WT and *h1* mutant. The "Regions that gain NRPE1 in h1" peaks are more enriched in heterochromatic regions, while " Regions that lose NRPE1 in h1" peaks are more enriched outside heterochromatic regions.

g. Figure 5: how many regions gain vs lose NRPE1 in h1 mutants? If the "redistribution causes loss" scenario applies, the numbers should overall be balanced but that does not seem the case. The loss case appears to be rather exceptional judging from the zigzagging meta-plot. Are these sites related to the sites taken over by PcG-mediated repression in h1 mutants?

As described in line 222 (previous version of the manuscript line 206), there are 15,075 sites that gain and 1,859 sites that lose NRPE1 in *h1*. Comparing these sites to

H3K27me3 in the Teano et al. study was an excellent suggestion. We compared sites that gain NRPE1 to sites that gain H3K27me3 in *h1*, finding a statistically significant overlap (2.4 fold enrichment over expected, hypergeometric test p-value 2.1e-71). Reciprocally, sites that lose NRPE1 were significantly enriched for overlap with H3K27me3 loss regions (1.6 fold over expected, hypergeometric test p-value 1.4e-4). This indicates that RdDM and H3K27me3 patterning are similarly modulated by H1. To directly test this, we reanalysed the H3K27me3 ChIP-seq data from Teano et al., finding coincident gain and loss of H3K27me3 at sites that gain and lose NRPE1 in *h1*. These results are described from line 250 and in Fig 6e, which supports a general role for H1 in preventing heterochromatin encroachment.

(6) Lines 166ff and Figure 3: The data walk towards the scenario of pathway redistribution but actually find that RdDM plays a minor role overall as a substantial increase in heterochromatin regions occurs in all contexts and is largely independent of RdDM.a. How exactly are DNA-methylation data converted across regions to reach a fraction score from 0 to 1? There is no explanation in the legend for the methods that allow to recapitulate.

We now explain our methods in full in the Materials and Methods and all the code for generating these has now been deposited on GitHub (https://github.com/Zhenhuiz/H1restricts-euchromatin-associated-methylation-pathways-from-heterochromaticencroachment). Briefly, BSMAP is used to calculate the number of reads that are methylated vs unmethylated on a per-cytosine basis across the genome. Next, the DNA methylation fraction in each region is calculated by adding all the methylation fractions per cytosine in a given window, and divided by the total number of cytosines in that same window (ie mC/(unmC+mC)) i.e. this is expressed as a fraction ranging from 0 to 1.

“0” indicates this region is not methylated, and “1” indicates this region is fully methylated (every cytosine is 100% methylated).

b. Kernel plots? These are slang for experts and should be better described. In addition, nothing is really concluded from these plots in the text, although they may be quite informative.

Kernel density plots show the proportion of TEs that gain or lose methylation in a particular mutant, rather than the overall average as depicted in the methylation metaplots above. We now describe the kernel density plots in more detail in the Figure 3 legend.

(7) Figure 4: This could be a very interesting analysis if the reader could actually understand it.a. The legend is minimal. What is the meaning of hypo and hyper regions indicated to the right of Figure 4c?b. The color scale represents observed/expected values. What exactly does this mean? Mutant vs WT?c. Some comparisons in 4a are cryptic, e.g. h1 nrpe1 nrpe1 vs CHH?d. Figure 4d focuses on a correlation square of relevance, but why? Interestingly the square does not correspond to any "hypo" or "hyper" label?

Thank you, we have revised Figure 4 and legend based on these suggestions to clarify all of the above.

(8) Lines 226 and Figure 6B. De novo (or increased) targeting of SUVH1 to heterochromatic sites in h1 mutants, similar to NRPE1, is used to support the argument that more access allows other chromatin modifiers to encroach. SUVH1 strongly depends on RdDM for its in vivo binding and may be the least conclusive factor to argue for a "general" encroachment mechanism.

We appreciate the reviewers point here. Something that is entirely independent of RdDM following the same pattern would be stronger evidence in favour of general encroachment. Excitingly, this is exactly what we provide evidence for when investigating the interrelationship with H3K27me3 and we appreciate the reviewer’s suggestion to check this! This data is now described in Figure 6e and line 250.

Minor:(1) Line 23: "Loss of H1 resulted in heterochromatic TE enrichment by NRPE1." This does not seem right. NRPE enrichment as TEs

Modified, (line 26) thank you.

(2) Lines 73-74: The idea that DDM1 displaces H1 in heterochromatic TEs is somewhat counterintuitive to model that heterochromatic TEs are unavailable for RdDM because of the presence of H1. Is this displacement non-permanent and directly linked to interaction with CMT2/3 Met1?

This is a very good question and we agree with the reviewer that the effect of DDM1 may only be transient or insufficient to allow for full RdDM assembly, or indeed there may be a direct interaction between DDM1 and CMTs/MET1. During preparation of these revisions, a structure of Arabidopsis nucleosome bound DDM1 was published, which provides some insight by showing that DDM1 promotes DNA sliding. This is at least consistent with the idea of DDM1 causing transient / non-permanent displacement of H1 that would be insufficient for RdDM establishment. We incorporate discussion of these ideas at line 80.

(3) Line 85: A bit more background on the Reader activator complex should be given. In fact, the reader may not really care that it was more recently discovered (not really recent btw) but what does it actually do?

We have quite extensively reconfigured this paragraph to take into account our new finding with H3K27me3, such that there is less emphasis on the reader activator complex. The sentence now reads as follows:

“We found that h1 antagonizes NRPE1 occupancy throughout the genome, particularly at heterochromatic regions. This effect was not limited to RdDM, similarly impacting both the methylation reader complex component, SUVH1 (Harris et al., 2018) and polycomb-mediated H3K27me3 (Teano et al., 2023).” (line 95).

Also, when describing the experiment the results section (line 241), we now provide more background on SUVH1’s function.

(4) Lines 80-81: Since it is already shown that RdDM associated small RNAs are more enriched in h1 at heterochromatin, help us to know what is precisely the added value of studying the enrichment of NRPE1 at these sites.

Good point. We have the following line: ‘...small RNAs are not a direct readout of functional RdDM activity and Pol IV dependent small RNAs are abundant in regions of the genome that do not require RdDM for methylation maintenance and that do not contain Pol V (Stroud et al., 2014).’ (line 90)

(5) Line 99: This seems to be the only time where the connection between long TEs and heterochromatic regions is mentioned but no source is cited.

We have added the following appropriate citations: (Bourguet et al., 2021; Zemach et al., 2013). (line 110).

(6) Line 100: DMRs is used for the first time here without explanation and full text. The abbreviation is introduced later in the text (Line 187).

Thank you, we now describe DMRs upon first use, line 112.

(7) Figure 2: Panels 2 c and d should show metaplots for WT and transgenes in one panel. There is something seriously wrong with the normalization in d or the scale for left and right panel is not the same. Neither legend nor methods describe how normalization was performed.

Thank you for pointing this out, the figure has been corrected. We have updated the Materials and Methods (line 365) and have added codes and pipelines to GitHub to explain the normalisation procedure in more detail (https://github.com/Zhenhuiz/H1restricts-euchromatin-associated-methylation-pathways-from-heterochromaticencroachment).